# How Mental Health Nurses Perceive the Implementation of Batho Pele Principles in a Selected Mental Health Hospital in Limpopo Province, South Africa

**DOI:** 10.3390/healthcare12232402

**Published:** 2024-11-29

**Authors:** Nkhensani Florence Mabunda

**Affiliations:** Department of Nursing Science, School of Health Care Sciences, Sefako Makgatho Health Sciences University, Molotlegi St, Ga-Rankuwa Zone 1, Ga-Rankuwa 0208, South Africa; nkhensani.mabunda@smu.ac.za; Tel.: +27-730-862-036

**Keywords:** implementation, Batho Pele principles, mental health nurses, mental health hospitals, service delivery, transformation

## Abstract

**Background:** Batho Pele is a South African legislative framework initiative introduced to improve the overall delivery of public services. The framework was introduced in 1997 and aimed at bringing a comprehensive transformation of the work ethics of all public servants, including health workers, at all levels of health hospitals. The study aims to investigate nurses’ perceptions of implementing Batho Pele principles in selected mental health hospitals. **Methods:** A quantitative descriptive survey design using a self-administered questionnaire was used to collect data. Probability-stratified random sampling was used to select the sample of 230 from the population of nurses. Statistical Package for the Social Sciences (SPSS) version 23 was used to analyze descriptive data. **Results:** The study showed that lack of knowledge, communication and practical skills, and human and material resources negatively affect the effective implementation of Batho Pele principles in the care of mental health service users. **Conclusions:** The Batho Pele principles are insufficiently implemented in mental health hospitals due to a shortage of staff to facilitate the implementation of Batho Pele principles. The researcher recommended an in-service program to train nurses in implementing Batho Pele principles to enhance the provision of exemplary mental health services.

## 1. Introduction

Batho Pele is a Sotho concept that translates to placing “people first” [1]. This concept is explicated in the literature as a framework to establish that citizens, regarded as customers of public services, are given primary consideration and treatment in the delivery of services [2]. Bearing this principle in mind, the citizen as a customer requires the service provider’s attentiveness to their perspectives and their inclusion in decision-making concerning service offerings [2,3]. Providing quality healthcare service is of concern worldwide. For example, the World Health Organization (WHO) has recommended an audit tool to assess the clinical performance of health services worldwide over a specific period [4]. In addition, the authors conveyed that the provision of quality health services is a South African ministerial priority that cannot be negotiated. In this regard, the authors indicated that the National Health Insurance (NHI) stipulates that compliance with National Core Standards (NCSs) has been established to improve health service quality. Moreover, implementing the Batho Pele framework is a significant aspect of improving the quality of healthcare services leaving the patient satisfied [5]. Implementation of the Batho Pele framework was established to promote responsibility and demonstrates a high predisposition for realizing the constitutional values to manage public administration in the country. This implies that all public employees are obligated to fulfill democratic rights by transforming and improving the service delivery so that it is aligned with Batho Pele principles by putting people first. The WHO (2019) report on the implementation of the Batho Pele principles shows that access to healthcare services is a major problem in several developing and underdeveloped countries. As noted in the recent literature, the WHO emphasized that quality healthcare services should be accessible to everyone including the MHCUs [6].

Nonetheless, challenges associated with non-adherence to Batho Pele principles were identified. Joel [7] identified that the implementation of the Batho Pele framework has been not much more than satisfactory. Additionally, moving public servants from “knowing” to “doing” minds was found to be the most challenging task in the implementation process. This revealed that implementation of Batho Pele principles did not result in a significant enough variation to serve its purpose because the implementers were not trained [8].

In the South African public health sector, Batho Pele principles, incorporated in the White Paper on Transforming Public Service Delivery (WPTPSD), are recognized as a policy framework aimed at making patients’ needs the central focus of healthcare service delivery. Implementing the Batho Pele policy remains a key priority for patient-centered care per the WPTPSD and the National Development Plan (NDP) [8]. The components of Batho Pele principles, as defined in the literature, and their applicability include consultation, service standards, access, courtesy, information, openness and transparency, redress, and value for money [9,10,11].

According to Mulaudzi et al. [12], South African nurses, as public service providers, are expected to adhere to the Public Service Act. In this regard, incorporating Batho Pele principles in daily nursing routines promotes high standards of professional ethics. This concurs with [11], which emphasized that nurses should prioritize patients before their own needs, as stated in the “Nurse’s Pledge of Service”. Moreover, the recent literature revealed that the Nurse’s Pledge of Service forces nurses to declare publicly that they will deliver quality healthcare services with respect and dignity without allowing nationality, race, or religion to intervene in their duties [13].

The Public Service Commission (PSC) conducted eight studies between 2005 and 2009 to measure how the public sector complied with implementing the Batho Pele policy. The findings of these studies revealed that almost all departments that responded to the study indicated that they are involved in the implementation of principles [14]. A study was conducted in 2010 in a Gauteng public hospital to detect the shortcomings in the application of Batho Pele principles [15]. The study findings revealed that some of the Batho Pele principles were implemented effectively. As a result, patients were not satisfied with the lack of management skills and knowledge of health professionals.

Additionally, the NCS tool was devised in this context, encompassing Batho Pele principles, to establish a shared understanding or standardized definition of quality care. Much has been implemented over the years to improve the quality of healthcare services. The literature, however, shows that South Africa still needs further interventions to improve the quality of healthcare services [16]. Another study highlighted that the Batho Pele framework was developed to enhance effective service delivery in public sectors [17].

Nevertheless, studies have been conducted regarding implementing Batho Pele principles in public and private hospitals. The authors also highlighted that the provision of quality healthcare services can be improved by assessing implementation of Batho Pele principles [18,19,20]. The authors further indicated that Performance Management Systems were established to monitor and evaluate whether healthcare services are provided effectively and correctly [18]. Batho Pele principles were introduced with the intention of being practiced continuously to assess whether scientific guidelines and policies are implemented, thus ensuring that patients are treated with human dignity and consideration [19]. In addition, the National Core Standards (NCSs) in South Africa were also introduced to improve the quality of healthcare delivery in public hospitals [20]. A study was also conducted in Limpopo province to describe the understanding of quality nursing care in a public hospital [20]. The study alluded that inadequate healthcare resources affect the provision of quality healthcare services that contribute to South African legal action and mortality. This same study highlighted the importance of the Department of Health in providing nurses with comprehensive resources to improve healthcare quality. Indeed, [20] argued that healthcare institutions with limited resources are a significant problem and a strategic plan is required to enhance the implementation of Batho Pele principles to improve service delivery.

The study by Maphumulo et al. [20] also highlighted that there is a need to improve nursing practice to evaluate how the Batho Pele framework is implemented to increase its efficacy. Other studies demonstrate that implementation of Batho Pele principles takes inspiration from the private sector to improve the quality of healthcare services [21,22]. A study conducted in KwaZulu Natal identified that evaluating the implementation of Batho Pele principles is a relevant action plan to improve the quality of service delivery [23]. A study conducted in North-West Psychiatric Hospital revealed that lack of knowledge is a problem that negatively affects the implementation of Batho Pele principles in mental health institutions [24]. The authors also recommended that further studies should be conducted to describe how health providers implement Batho Pele principles in other provinces. The fact that the authors conducted the study in the selected hospital prompted the researcher to believe that there is inadequate information to link the provision of public services and the application of Batho Pele principles specifically in mental health institutions (MHHs). Within this context, little is known about the perspectives of nurses regarding the implementation of Batho Pele principles in mental health institutions in Limpopo province. Therefore, this article attempts to describe nurses’ perceptions of implementing Batho Pele principles to maintain and facilitate effective nursing care in some selected mental health hospitals.

The researcher worked in a mental health hospital where nurses expressed their concern about how Batho Pele principles were introduced to improve the quality of the provision of healthcare services. A qualitative study to describe nurses’ perceptions of the developed strategies to facilitate effective nursing care indicated that the provision of effective care needs to be strengthened through the use of developing strategies that will improve the quality of nursing practice [25]. In this study, the researcher noted that the evaluation process for implementing Batho Pele principles in the mental health institution in monthly audits may be inadequate. The Batho Pele principles are pasted on the walls in every ward/unit. The auditing tool includes statements that assess whether the ward/unit complies with implementing Batho Pele principles to the patients. However, the tool does not clarify how to assess the implementation of Batho Pele principles at all. In addition, it is noted in the literature that public healthcare services are faced with repeated complaints regarding inadequate and unequal access to health service delivery [26].

Moungui et al. [27] alluded that posters are appropriate for disseminating health information. In the current study, the presence of a Batho Pele principles poster on the ward walls seems to be used as a sign of compliance with implementing the principles. Challenges arose in implementing Batho Pele principles when admitting MHCUs who were hyperactive or had an intellectual disability because of their distinct conditions. It was suggested in the literature that all healthcare providers must be equipped with adequate competencies to provide quality mental health services regardless of mental health disorders [27]. A study by Molefe [28] found that nurses caring for hyperactive individuals and those with an intellectual disability experience psychological burdens linked to stress and emotional trauma. Moreover, Hadebe [29] reported that the quality of mental health services remains less of a public health priority, especially in hyperactive and intellectual disability wards, resulting in limited access to healthcare services due to a lack of skills to implement service standards. This indicates that simply adopting Batho Pele principles may not be enough in this circumstance. Therefore, this study was designed to describe how mental health nurses perceive the implementation of Batho Pele principles in selected MHHs.

## 2. Materials and Methods

A quantitative descriptive research design was used to examine a phenomenon of interest at the selected MHHs. Batho Pele principles are widely understood within South Africa. It is a South African political framework aimed to improve the quality and accessibility of public services. The term “Batho Pele” is a Sesotho concept that means “People First” and is used to encourage public servants to be committed to prioritizing people over their own needs and to find ways to improve service delivery. The initiative was launched in 1997 by the Mandela administration to address the shortcomings of public service in a newly democratic South Africa [2,8,26,30]. This study was conducted in the clinical settings of hospitals in the Mopani, Vhembe, and Capricorn districts of the Limpopo province in South Africa. There is only one mental health hospital (MHH) in the Greater Giyani municipality in the Mopani district, one in the Greater Thulamela municipality in the Vhembe district, and one in the Lepelle-Nkumpi local municipality in the Capricorn district. All community MHHs in Limpopo in the mentioned districts admit both long-term and acute psychiatric MHCUs. In addition, mental health services are also offered in primary healthcare settings including mobile clinics.

The accessible target population was nurses working at the selected MHHs. In this study, the probability-stratified random sampling method was used because each participant in the population has an equal chance of being included in the sample [31,32]. The respondents who met the inclusion criteria were all nurses on duty during data collection, both male and female of all ages, who were willing to participate in the study voluntarily. The population was divided into strata or sub-groups as follows: A—operational managers, B—registered nurses, C—enrolled nurses, and D—enrolled nursing auxiliaries. A total of 230 respondents were drawn randomly from 271 nurses to achieve a greater degree of representativeness. The population was stratified according to any number of attributes, such as age, gender, years of service, and level of qualification.

### 2.1. Data Collection

Data collection occurred after receiving permission to collect data from respective authorities. An appointment was made with the nursing manager and operational manager a week before data collection. The data collection method for this research project was self-administered questionnaires that were handed to the respondents who were on duty. The researcher distributed questionnaires to the respondents. Each respondent completed a questionnaire on their own without discussing it with others. The researcher was available in case problems were experienced. The questionnaire consisted of two sections: Section 3.1, demographic distribution, and Section 3.2, nurses’ perceptions of implementing Batho Pele principles.

The types of questions which the author selected include 23 closed-ended questions and one open-ended question. Close-ended questions allow the respondents to select one response from the number provided according to the instructions. One open-ended question, “What is your suggestion/comment on the overall implementation of Batho Pele principles?”, was included to provide decisive information that closed-ended questions cannot deliver [33]. Respondents were given the opportunity to comment [34] on how they implement Batho Pele principles. The researcher ensures validity and reliability by creating valid and reliable questions that address your research objective [35]. In this study, the researcher ensures that the information in the questionnaire is reliable and valid to answer how mental health nurses perceive implementing Batho Pele principles in the selected MHHs. Regarding questionnaire development, the author reviewed the literature to obtain information about nurses’ perceptions of implementing Batho Pele principles to develop a questionnaire. The draft questionnaire was sent to mental health experts to critique logical patterns of answers and whether the questionnaire will consistently measure what it purports to measure [36,37].

The author consulted a statistician to review and confirm whether the instrument would measure what it was intended to measure to enhance the quality of the study while developing the questionnaire. The researcher developed a four-response-option Likert-scale matrix (strongly agree, agree, disagree and strongly disagree) in consultation with the statistician to measure responses in frequency and percentage to calculate the scores. Moreover, a dichotomous scale (two-point scale) was used to collapse data into agree and disagree to block respondents’ opportunity to be neutral to a question [38]. The instrument was pre-tested by administering the questionnaire to 20 respondents who were not included in the main study. The purpose was to check the relevance and clarity of the questions as well as the time needed to complete the questionnaire.

### 2.2. Data Analysis

Data were analyzed statistically by a statistician using the Statistical Package for Social Sciences (SPSS) version 23 program. The questionnaire comprises demographic data and 23 self-assessment questions for respondents’ perceptions concerning the implementation of Batho Pele principles; frequencies and percentages were determined and attached. The numerical percentages for organizing and interpreting data were used to enable trends and differences to be noted and calculations of the simple statistics such as frequency, percentage, and proportion of the score. The findings of this study were interpreted and discussed from questionnaires with references to the literature review where applicable.

### 2.3. Ethical Considerations

Permission to conduct the current study was obtained from the University of Venda’s Institutional Review Board (IRB) (Ref: SHS/12/PDC/11/1012). The ethical clearance certificate was attached to apply to the Limpopo Department of Health to collect data at the MHHs in the Mopani, Vhembe, and Capricorn districts. Ethical clearance and approval letters from the Limpopo Department of Health and the respective districts were attached to the applications to the MHHs. Permission to conduct was also obtained from the chief executive officers of the selected MHIs. Arrangements were made with the nursing managers to access the participants. The researcher observed the right to privacy, autonomy, and self-determination through voluntary participation by means of informed consent. The respondents were instructed not to put their names in the questionnaire and also reassured that all information would be treated confidentially.

## 3. Results

The response rates of the self-administered questionnaires were distributed to the respondents; it was found that of 250 questionnaires, 243 were returned at a rate of 97%. Of the 243 questionnaires collected, 230 were operational for the statistical data analysis, and 13 questionnaires were not captured due to incomplete data or multiple answers. The questionnaire consisted of two sections: Section 3.1 collects the demographic data of respondents, as indicated in Table 1, and Section 3.2 consists of self-assessment questions to describe nurses’ perceptions of implementing Batho Pele principles in selected MHHs in Table 2 and Table 3.

### 3.1. Demographic Profiles of the Respondents

Table 1 shows the distribution of nurses, revealing that the majority of 67% (165) respondents who worked for more than 11-25 years of service have experience in providing mental health care services. The majority of respondents 43% (99) were above 40 years which shows that more adults were dominating the mental health services. For gender, the majority 68% (156) of respondents were females 30% (69) dominating. Regarding their category, 34% (79) were registered nurses, 22% (51) were enrolled nurses and the majority, 41% (95), were enrolled nursing axillaries.

### 3.2. Nurses’ Perceptions of Implementing Batho Pele Principles

Data were gathered from the questionnaire subject to the number of respondents, of which the subjects’ responses for each question were added together to find the highest numbers and were then presented in percentages in tabular form, as shown in Table 2 below.

Table 2 shows respondents’ perceptions of the implementation of Batho Pele principles to measure statements of agreement. The 4-point Likert rating scale was used to measure responses from the respondents, which were given as percentages [35,36]. The Likert scale was used to investigate nurses’ perceptions of the implementation of Batho Pele principles at the selected MHHs. In addition, a dichotomous scale was used to collapse data into a two-point scale to prevent respondents from being neutral to a question [37,38].

## 4. Discussion

This section summarizes the study findings from the respondents who were on duty during data collection. Demographic profiles of the respondents encompass year of service, age, as indicated in Table 1, and gender, as well as category of nurse. This section aimed to assess how mental health nurses implement Batho Pele principles at the selected MHHs. The discussion of findings was based on the results presented concerning the literature review. In addition, some items with similar focus were clustered together for easy interpretation of the meanings. These include approaches to improve the quality of mental health care; implementation of the principles; and knowledge about the application of the principles. Moreover, respondents were allowed to suggest or comment on the implementation of Batho Pele principles.

*Approaches to improve the quality of mental health care services*. Study findings show that the majority of the respondents (81%) agreed that Batho Pele principles are displayed in the unit/ward, prompting them to be committed to their daily activities [39,40,41,42,43]. Of the 230 respondents, 81% agreed that a complaints procedure brochure is displayed and that there is a suggestion box for service users to drop their complaints in the hospital. These results show that nurses understand the significance of handling complaints when standards of service fall below the promised level [44]. Of the 230 respondents, 84% disagreed that Batho Pele principles are included in the in-service education program in their wards/units. This finding indicates that the in-service training of nurses should be included to upskill nurses and expand their capabilities to render quality healthcare services [45]. This revealed that nurses are aware of the importance of the feedback from the patient satisfaction survey in assessing patient satisfaction with nursing services [46] and when planning interventions to meet MHCUs’ needs [47].

*The implementation of Batho Pele principles*. The current study shows that 61% of the 230 respondents agreed that MHCUs are treated as customers. This implies that nursing staff can listen to MHCUs’ views and take them into account when making decisions about what mental health services should be provided [45]. The findings of the current study show that 62% of the 230 respondents agreed that MHCUs are treated with courtesy and consideration. This implies that nurses consider the foundational values—to treat MHCUs with dignity, compassion, and kindness—in all interactions, which shows that courtesy is a significant aspect of nursing practice [46].

*Knowledge of the application of Batho Pele principles*. The study findings show that some of the nurses know how Batho Pele principles are implemented. However, respondents disagreed that nurses have sufficient knowledge, which shows the great challenge of implementation of service improvement, which might be due to insufficient knowledge [47]. This implies that nurses should have sufficient knowledge and practical skills to ensure that patients receive equal treatment regardless of conditions [48].

*Suggestions or comments on the implementation of Batho Pele principles.* The suggestions or comments raised by the participants in Table 3 imply that participants know the implementation of Batho Pele principles as well as what could be implemented to improve service provision and the quality of life of the customers. The study findings highlighted the need to support nurses to strengthen the implementation of Batho Pele principles in the MHHs [49,50,51].

## 5. Conclusions

This study was conducted to investigate nurses’ perceptions of implementing Batho Pele principles in selected MHHs as a strategy to improve the quality of nursing care. Study findings revealed that the in-service education program is significant in equipping nurses with knowledge about the implementation of Batho Pele principles. In addition, the provision of quality mental health services may also improve. The author hopes that this study will prompt hospital management and provincial and national departments to initiate programs to ensure the effective implementation of Batho Pele principles in nursing practice.

## 6. Limitations

Approval from the provincial Department of Health was delayed, which affected the progress of the study. Approval from one hospital was also delayed; the new chief executive officer was on orientation, and the nursing service manager was on leave, which also slowed down the progress of the study. In addition, the study targeted the nurses working at the MHHs, so was limited to nurses working in mental health units within public hospitals in the same districts.

## Figures and Tables

**Table 1 healthcare-12-02402-t001:** Demographic distribution.

Demographic Distribution	Characteristics	Number of Respondents	Percentages
Years of service	<10	60	26
	11–15	42	18
	16–24	85	37
	>25	38	12
Age	20–30	57	25
	31–40	56	24
	41–50	76	33
	>50	36	16
Nursing category	Enrolled nursing auxiliary	95	41
	Enrolled nurse	51	22
	Registered nurse	79	34
	Operational managers	9	4
Gender	Male	69	30
	Female	156	68

**Table 2 healthcare-12-02402-t002:** Respondents’ perceptions concerning the implementation of Batho Pele principles: frequencies and percentages.

Responses	Agree		Disagree	
Statement				
Approach to Improve the Quality of Mental Health Care	Number of Respondents	%	Number of Respondents	%
1. Batho Pele principles are displayed in the unit/ward	187	81	43	19
2. Batho Pele principles are included in the in-service education program in your unit	37	16	193	84
3. The hospital displayed the complaints procedure brochure and suggestion box	187	81	43	19
4. The hospital has a healthcare service delivery improvement plan to address complaints	141	61	89	39
5. The hospital constantly trains nurses on Batho Pele principles	122	53	108	47
6. The hospital persistently conducts patient satisfaction surveys	38	17	192	83
7. The promised level and quality of services are always of the highest standard	112	49	118	51
8. The hospital responds swiftly and sympathetically when standards of services fall below the promised standards	111	48	119	52
9. The hospital developed policies and protocols to provide quality mental health care services or integrate them into existing framework	119	52	111	48
10. The hospital set simple, measurable, accessible, and realistic (SMART) service standards of quality that MHCUs receive	127	55	103	42
11. The hospital supports the needs of MHCUs	110	49	118	51
12. Services rendered encourage innovation and reward excellence	115	50	115	50
13. The hospital endeavors to assess the impact of mental health services annually to ascertain whether it is achieving specified objectives	56	76	174	24
14. Managers lead by example and endeavor to ensure that the vision, mission, and goals are articulated and embraced to maintain leadership and strategic direction	111	48	119	52
**Implementation of the principles**	**F**	**%**	**F**	**%**
15. MHCUs are treated as customers, listening to their views, and taking them into account in making decisions about what mental health services should be provided	140	61	90	39
16. MHCUs are consulted about the level and quality of mental health services they receive and, wherever possible, given a choice about services that are offered	112	49	118	51
17. MHCUs have equal access to all mental health services rendered	132	57	98	43
18. MHCUs are treated politely and with a friendly disposition with courtesy and consideration	140	62	88	38
19. MHCUs are given full, accurate information of mental health services rendered	127	55	103	45
**Knowledge of the application of the principles**	**F**	**%**	**F**	**%**
20. Nurses have knowledge of how Batho Pele principles are implemented	58	22	172	78
21. Nurses recognize that openness and transparency are the cornerstones of our daily activities	63	27	167	73
22. Nurses respect the right of MHCUs to complain if our services are poor or unsatisfactory	55	24	175	76
23. Nurses endeavor to use health resources efficiently, effectively, and economically to comply with value for money	113	49	117	51

**Table 3 healthcare-12-02402-t003:** Suggestions/comments on the overall implementation of Batho Pele principles.

Comments		
What Is Your Comment on the Overall Implementation of Batho Pele Principles?	F	%
1. Regular in-service training about the implementation of the Batho Pele principles	8	18
2. Lack of human and material resources to improve the services	13	30
3. Advertising vacant posts of operational managers to facilitate the implementation process	3	8

## Data Availability

Data and materials are available on request. The email of the corresponding author is nkhensani.mabunda@smu.ac.za.

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
