# Peer review of "How Mental Health Nurses Perceive the Implementation of Batho Pele Principles in a Selected Mental Health Hospital in Limpopo Province, South Africa"

_healthcare, 2024, doi:10.3390/healthcare12232402_

Round 1
Reviewer 1 Report
Comments and Suggestions for Authors
Dear Editor,
Thank you for the invitation to review this manuscript. Please find our comments for the authors below.
- Do you have the inclusion criteria for your sample? If yes, please add the information.
- Need more information about the instrument/questionnaire used to collect the data. What questions were asked both in open-ended and closed-ended questions in the questionnaire? How many questions? And how to calculate the scores? Please add more information.
- Please add one section for data analysis under “materials and methods” section.
- For demographic characteristics, I suggest the authors combine all demographic characteristics into one table.
- Page 5 line 184, “of 220 nurses,” it should be “of 230 nurses.”
- Page 5 lines 185-186, “age from 40 to 50 years” should be changed to “age from 41 to 50 years.” Make it consistent with the table that is referred to.
- Page 5 lines 186-187, it is written that “majority of respondents 43% (99) were above 40 years.” However, in Table 2, respondents aged above 40 years count for 49% (112). Please clarify.
- Page 5, lines 195–200, that paragraph explains about data analysis. I suggest the authors move that paragraph to the data analysis section in the method part.
- Page 9 lines 334-338 (discussion section) about the number of nurses suggesting the implementation of Batho Pele principles; in which table can we see those results? Please explain.
Comments on the Quality of English Language
- There are many writing errors that need to be fixed, for example,
- In the abstract, lines 12–13, it is written, “To assess how mental health nurses implement Batho-Pele principles 12 in selected mental health hospitals.” Please revise this sentence into a complete sentence: “The aim of this study is to assess how mental health nurses implement Batho-Pele principles 12 in selected mental health hospitals.”
- Page 8, line 285. “In this regard41 highlighted that managers should be …” What is that 41? Is that a reference? If yes, it should be written in the bracket (). Please revise.
- Page 8 line 289, “Implementation of the Batho Pele principles.” This statement cannot be a sentence. I assume this is a sub-heading. If yes, please write it in a way that it will differ from other sentences. Please make the font bold or italicized.
- The same comment for the statement “Knowledge of the application of the Batho Pele principles.” On page 9, line 317, and “Suggestions or comments on the implementation of the Batho Pele principles.” On page 9, line 334. Please bold the font if this is a sub-heading of the discussion section.
- Page 8 line 2, it is written, “WHO emphsised.” Do you mean “emphasized”? Please revise.
- Page 8 line 293, “According to9 since South Africa 1994 there have been service delivery protests …” This sentence is difficult to understand. What does it mean “since South Africa 1994? Also, what is 9 after “according to”? Is that a reference or only a typo? Please clarify.
- On page 8, line 300, it is written, “MHCUs to come first46 as indicated." Again, what does first46 mean?
- Please pay attention to the in-text citation and follow the correct way to insert the reference. The reference in the bracket should be inserted before the full stop. Writing it after the full stop will confuse the reader whether the reference belongs to the previous or the next sentence. For example, on page 2 lines 82-85, it is written, “[21] A study conducted in Kwazulu Natal identified that evaluating the implementation of the Batho Pele principles is a relevant action plan to improve the quality of healthcare service delivery. [22] A study conducted in Nort-West Psychiatric Hospital revealed that …” It is not clear to which sentences do the references numbers 21 and 22 belong.
- Also, on page 9, line 322, it is written, “[31] in addition, [47] alluded that nurses are expected to apply ethical …” It is not clear if 31 belongs to this sentence or the previous sentence. Please correct the in-text citation way for the entire manuscript.
Author Response
REVIEWER 1
Comment 1. Thank you for reviewing this manuscript. Please find our comments from the authors below.
Comment 2. Do you have the inclusion criteria for your sample? If yes, please add the information.
Page 3, Line 160-162: Yes, the respondents who met the inclusion criteria were all nurses who were on duty during data collection, both male and female of all ages, who were willing to participate in the study voluntarily.
Comment 3. Need more information about the instrument/questionnaire used to collect the data. What questions were asked both in open-ended and closed-ended questions in the questionnaire? How many questions?
Page 3. Line 176: The types of questions the author selected include 23 closed-ended and one open-ended question.
Page 3. Line 179-180: One open-ended question “What is your suggestion/comment on the overall implementation of Batho Pele principles?” was included….
And how to calculate the scores? Please add more information.
More information was added as indicated below.
Page 4. Line 191-197: The author consulted a statistician to review and confirm whether the instrument would measure what it was intended to measure to enhance the quality of the study while developing the questionnaire. The researcher developed a four response options Likert scale matrix (strongly agree, agree, disagree and strongly disagree) in consultation with the statistician to measure responses in frequency and percentage to calculate the scores. Moreover, a dichotomous scale (two-point scale) was used to collapse data into agree and disagree to block respondents’ opportunity to be neutral to a question. [29]
Comment 4. Please add one section for data analysis under “materials and methods” section.
Page 4. Line 101-110: Data were analyzed using the Statistical Package for the Social Science (SPSS) version 23. The questionnaire comprises demographic data and 17 self-assessment questions for respondents' perception concerning the implementation of Batho Pele principles frequency and percentages were developed and attached. The study results were presented in the form of tables in frequency and percentages. Additionally, the results were interpreted and discussed with relevant references to the literature review where applicable.
Comment 5. For demographic characteristics, I suggest the authors combine all demographic characteristics into one table.
Page 5. Line 236: demographic characteristics combined into one table
Comment 6. Page 5 line 184, “of 220 nurses,” it should be “of 230 nurses.”
Page 6 The paragraph was removed as recommended by the second reviewer.
Comment 7 Page 5 lines 185-186, “age from 40 to 50 years” should be changed to “age from 41 to 50 years.” Make it consistent with the table that is referred to.
Page 6 The paragraph was removed as recommended by the second reviewer
Comment 8. Page 5 lines 186-187, it is written that “majority of respondents 43% (99) were above 40 years.” However, in Table 2, respondents aged above 40 years count for 49% (112). Please clarify.
Page 6 The paragraph was removed as recommended by the second reviewer
Comment 9. Page 5, lines 195–200, that paragraph explains about data analysis. I suggest the authors move that paragraph to the data analysis section in the method part.
Page 5. Line 202-210: data analysis moved to a section in the method
Comment 10. Page 9 lines 334-338 (discussion section) about the number of nurses suggesting the implementation of Batho Pele principles; in which table can we see those results? Please explain.
Page 8. Line 224-226: Table 3 Suggestion/comment on the overall implementation of Batho Pele principles. Added.
- There are many writing errors that need to be fixed, for example,
Comment 1. In the abstract, lines 12–13, it is written, “To assess how mental health nurses implement Batho-Pele principles 12 in selected mental health hospitals.” Please revise this sentence into a complete sentence: “The aim of this study is to assess how mental health nurses implement Batho-Pele principles 12 in selected mental health hospitals.”
Page 2. Line 12: “The study aims to assess” written
Comment 2. Page 8, line 285. “In this regard41 highlighted that managers should be …” What is that 41? Is that a reference? If yes, it should be written in the bracket (). Please revise.
Page 9. Line 274: [41] is written in brackets
Comment 3. Page 8 line 289, “Implementation of the Batho Pele principles.” This statement cannot be a sentence. I assume this is a sub-heading. If yes, please write it in a way that it will differ from other sentences. Please make the font bold or italicized.
Page 9. Line 177: The implementation of the Batho Pele principles. Italicized.
Comment 4. The same comment for the statement “Knowledge of the application of the Batho Pele principles.” On page 9, line 317, and “Suggestions or comments on the implementation of the Batho Pele principles.” On page 9, line 334. Please bold the font if this is a sub-heading of the discussion section.
Page 9. Line 285: Knowledge of the application of the Batho Pele principles. Italicized.
Comment 5. Page 8 line 2, it is written, “WHO emphasized.” Do you mean “emphasized”? Please revise.
Page 9. Removed as recommended by the second author
Comment 6. Page 8 line 293, “According to9 since South Africa 1994 there have been service delivery protests …” This sentence is difficult to understand. What does it mean “since South Africa 1994? Also, what is 9 after “according to”? Is that a reference or only a typo? Please clarify.
Page 9. Removed as recommended by the second author
Comment 7. On page 8, line 300, it is written, “MHCUs to come first46 as indicated." Again, what does first46 mean?
Page 9. Line 316-317: Removed as recommended by the second author
Comment 8. Please pay attention to the in-text citation and follow the correct way to insert the reference. The reference in the bracket should be inserted before the full stop. Writing it after the full stop will confuse the reader whether the reference belongs to the previous or the next sentence. For example, on page 2 lines 82-85, it is written, “[21] A study conducted in Kwazulu Natal identified that evaluating the implementation of the Batho Pele principles is a relevant action plan to improve the quality of healthcare service delivery. [22] A study conducted in Nort-West Psychiatric Hospital revealed that …” It is not clear to which sentences do the references numbers 21 and 22 belong.
Page 9. Line 340: The in-text citation was corrected throughout the document.
Comment 9. Also, on page 9, line 322, it is written, “[31] in addition, [47] alluded that nurses are expected to apply ethical …” It is not clear if 31 belongs to this sentence or the previous sentence. Please correct the in-text citation way for the entire manuscript.
The in-text citation was corrected throughout the document.

Reviewer 2 Report
Comments and Suggestions for Authors
I would like to express my sincere gratitude for the opportunity to review this valuable research, which evaluates how psychiatric nurses in South Africa implement the Batho Pele principles. This study offers significant implications for nursing practice in psychiatric hospitals in South Africa. However, this manuscript requires substantial revisions.
Overall Comments: This study investigates nurses' perceptions of implementing the Batho Pele principles and presents a descriptive analysis. Given the absence of hypothesis testing or in-depth analysis, classifying this as an "Original Article" might be inappropriate. Perhaps "Case Report" or "Preliminary Communication" would be more suitable categories.
Abstract: The abstract should concisely state the study's objectives, methods, analysis, results, and discussion. While a brief overview of the Batho Pele principles is necessary for context, the primary focus should be on the research's core findings.
1. Introduction
The introduction should clearly articulate the challenges associated with non-adherence to the Batho Pele principles in South Africa, particularly their impact on patient outcomes, supported by relevant literature.
You should provide a comprehensive background on the significance of the Batho Pele principles in the South African healthcare context, especially in the realm of mental health, drawing on studies from other countries.
Given the existing body of research on the Batho Pele principles, a more robust justification is needed for conducting another survey among psychiatric nurses regarding their implementation of these principles. The literature should be used to explain why this specific research question is still relevant.
The manuscript exhibits a noticeable lack of citations in several places. Adequate citations are crucial for establishing the credibility of your research and demonstrating its connection to previous studies.
Lines 27-30: This concept is explicated in the literature as a framework to establish that citizens, as customers of public services, are given primary consideration. Please provide a citation to support the claim that this concept is a framework to establish that citizens, as customers of public services, are given primary consideration.
・ Lines 27−30: This concept is explicated・・・, Please provide a citation to support the claim that this concept is a framework to establish that citizens, as customers of public services, are given primary consideration.
・ The phrase "According to [number]" is unconventional. Please replace it with the author's name, e.g., "According to [author's name] and colleagues...". Apply this change consistently throughout the paper.
・ Lines 88−96: The rationale for this study could be strengthened by citing relevant literature in the first half of this paragraph.
・ Lines 97―101: While the researcher's personal observations are insightful, the introduction should primarily rely on existing literature to establish a strong foundation for the research. Please incorporate qualitative data, literature review results, and the researcher's experiences to provide a more comprehensive and objective argument.
・ Lines 106−107: Please support the claim that challenges arose in admitting Hyperactive and Intellectual Disability MHCUs with relevant citations.
Materials and Methods
While the Batho Pele principles may be widely understood within South Africa, international readers may be less familiar with them. Please provide a concise explanation of these principles in the Methods section to ensure clarity for all readers.
If you choose to include the specific name of the hospital in lines 113-122, please provide additional context, such as a brief description or a relevant URL, to help readers better understand the institution and its role in this study.
2.1 Data collection
・ Please provide separate subsections for "survey contents" and "analysis methods" to offer a detailed explanation of the data collection process.
・ The ethical approval process should be discussed in the "ethical considerations" section. Please limit the information in lines 132-134 to the data collection methods.
・ Lines135−137: Please provide specific details about the distribution method of the self-administered questionnaires.
・ Given that the questionnaire was developed by the researcher, please describe the questionnaire development process.
2.1 Ethical considerations
Lines 152-153: You mention obtaining permission from the University of Venda's Higher Degrees Committee (Ref: SHS/12/PDC/11/1012). Can you clarify if this committee functions as an Institutional Review Board (IRB)? Given that ethical considerations were mentioned in the data collection section, it would be helpful to elaborate on the ethical approval process in this section.
3. Results
・ Table 1, Lines 176-180, Table 2, Lines 184-194: The information presented in these tables is redundant with the text. Please remove the redundancy and present the information in either the tables or the text, but not both.
・ Lines 195-200: The paragraph starting with "Data was analyzed statistically by..." appears to describe the analysis methods rather than the results. Please move this information to the methods section.
・ Lines 202-203, Lines 218-220: The objectives of the study are repeated in these lines. Please delete these repetitions.
・ Table 3: If "F" represents frequency, there seems to be an inconsistency between the Likert scale (which typically has four categories: strongly agree, agree, disagree, strongly disagree) and the presented data. Please clarify if "F" represents the number of respondents rather than frequency. Additionally, the order of the Likert scale categories in Table 3 should generally be "strongly agree, agree, disagree, strongly disagree." Furthermore, the binary variables "agree" and "disagree" in the lower part of the table are unclear. Please remove this row if the corresponding values are not provided.
4. Discussion, 5. Conclusion
Given the relatively simple nature of the data analysis involving primarily the tabulation of responses, the discussion section appears excessively lengthy. Additionally, the discussion seems to rely heavily on existing literature rather than focusing on the implications of the current findings. The discussion should be centered around the results of this study. It is recommended to concisely summarize the most significant findings from the data analysis.
Furthermore, after revising the discussion, the conclusion should briefly address the potential clinical applications of this research and its potential contributions to nursing practice and patient care.
Literature citations
Please place all reference numbers within the period at the end of the sentence. For example, "This finding is consistent with previous research [1]."
Author Response
REVIEWER 2
- Overall Comments: This study investigates nurses' perceptions of implementing the Batho Pele principles and presents a descriptive analysis. Given the absence of hypothesis testing or in-depth analysis, classifying this as an "Original Article" might be inappropriate. Perhaps "Case Report" or "Preliminary Communication" would be more suitable categories.
Comment 1. Abstract: The abstract should concisely state the study's objectives, methods, analysis, results, and discussion. While a brief overview of the Batho Pele principles is necessary for context, the primary focus should be on the research's core findings.
Page 1. Line 12: The study aims to investigate nurses' perceptions of implementing the Batho-Pele principles in selected mental health hospitals.
Page 3. 107-108 122-123 Therefore, this study was designed to describe nurses' perceptions of implementing Batho-Pele principles in selected MHHs.
- Introduction
Comment 2. The introduction should clearly articulate the challenges associated with non-adherence to the Batho Pele principles in South Africa, particularly their impact on patient outcomes, supported by relevant literature.
You should provide a comprehensive background on the significance of the Batho Pele principles in the South African healthcare context, especially in the realm of mental health, drawing on studies from other countries.
Page 1-2. Line 35-48 information to address the above comment added
Given the existing body of research on the Batho Pele principles, a more robust justification is needed for conducting another survey among psychiatric nurses regarding their implementation of these principles. The literature should be used to explain why this specific research question is still relevant.
Page 3. Line 103-106 justification added
The manuscript exhibits a noticeable lack of citations in several places. Adequate citations are crucial for establishing the credibility of your research and demonstrating its connection to previous studies.
Citations provided in the introduction
Comment 3. Lines 27-30: This concept is explicated in the literature as a framework to establish that citizens, as customers of public services, are given primary consideration. Please provide a citation to support the claim that this concept is a framework to establish that citizens, as customers of public services, are given primary consideration.
Page 1. Line 30, citation provided
Ngidi et al. Nyelisani et al. Maphumulo et al.
Comment 4. ・ Lines 27−30: This concept is explicated・・・, Please provide a citation to support the claim that this concept is a framework to establish that citizens, as customers of public services, are given primary consideration.
Page 1. Line 60, citation provided
Comment 5. ・ The phrase "According to [number]" is unconventional. Please replace it with the author's name, e.g., "According to [author's name] and colleagues...". Apply this change consistently throughout the paper.
Page 2. Line 57, Mulaudzi et al. [10], added
Comment 6. ・ Lines 88−96: The rationale for this study could be strengthened by citing relevant literature in the first half of this paragraph.
Page 3. Line 115-118
Comment 7. ・ Lines 97―101: While the researcher's personal observations are insightful, the introduction should primarily rely on existing literature to establish a strong foundation for the research. Please incorporate qualitative data, literature review results, and the researcher's experiences to provide a more comprehensive and objective argument.
Page 2. Line 123-126, 130-132 information added
Comment 8. ・ Lines 106−107: Please support the claim that challenges arose in admitting Hyperactive and Intellectual Disability MHCUs with relevant citations.
Page 2. Line 132-137 information provided
Comment 9. Materials and Methods
While the Batho Pele principles may be widely understood within South Africa, international readers may be less familiar with them. Please provide a concise explanation of these principles in the Methods section to ensure clarity for all readers.
Page 3. Line 144-149, A concise explanation of Batho Pele principles written.
Comment 10. If you choose to include the specific name of the hospital in lines 113-122, please provide additional context, such as a brief description or a relevant URL, to help readers better understand the institution and its role in this study.
Page 4. Line 152-154, Names of the hospital removed
Comment 11. 2.1 Data collection
・ Please provide separate subsections for "survey contents" and "analysis methods" to offer a detailed explanation of the data collection process.
Page 4. Line 174-176, information about the data collection process was added.
Comment 12. ・ The ethical approval process should be discussed in the "ethical considerations" section. Please limit the information in lines 132-134 to the data collection methods.
Page 5. Line: 203-211, Data analysis added
Comment 13. ・ Lines 135-137: Please provide specific details about the distribution method of the self-administered questionnaires.
Page 4. Line 172-173, information about the distribution method of the self-administered questionnaires. was added
Comment 14. ・ Given that the questionnaire was developed by the researcher, please describe the questionnaire development process.
Page 4. Line 188-192 information about the questionnaire development process written.
Comment 15. 2.1 Ethical considerations
Lines 152-153: You mention obtaining permission from the University of Venda's Higher Degrees Committee (Ref: SHS/12/PDC/11/1012). Can you clarify if this committee functions as an Institutional Review Board (IRB)? Given that ethical considerations were mentioned in the data collection section, it would be helpful to elaborate on the ethical approval process in this section.
Page 5. Line 214-220, Ethical consideration elaborated and removed from data collectionline:165-166.
Comment 16. 3. Results
・ Table 1, Lines 176-180, Table 2, Lines 184-194: The information presented in these tables is redundant with the text. Please remove the redundancy and present the information in either the tables or the text, but not both.
Page 5. Line 328 Demographic profiles of the respondents were combined into one table as recommended by the first authorand the description removed.
Comment 17・ Lines 195-200: The paragraph starting with "Data was analyzed statistically by..." appears to describe the analysis methods rather than the results. Please move this information to the methods section.
Removed to the methods section line 203-212.
Comment 18・ Lines 202-203, Lines 218-220: The objectives of the study are repeated in these lines. Please delete these repetitions.
The objectives of the study
Comment 19・ Table 3: If "F" represents frequency, there seems to be an inconsistency between the Likert scale (which typically has four categories: strongly agree, agree, disagree, strongly disagree) and the presented data. Please clarify if "F" represents the number of respondents rather than frequency. Additionally, the order of the Likert scale categories in Table 3 should generally be "strongly agree, agree, disagree, strongly disagree." Furthermore, the binary variables "agree" and "disagree" in the lower part of the table are unclear. Please remove this row if the corresponding values are not provided.
Page 5. Line 243 number of respondents written instead of “F”
The binary variables "agree" and "disagree" in the lower part of the table are explained on Page 4. Line 194-195 a dichotomous scale (two-point scale) was used to collapse data into agree and disagree to block respondents’ opportunity to be neutral to a question [32].
Therefore, strongly agree, agree, disagree, strongly disagree, removed
Comment 20. 4. Discussion, 5. Conclusion
Given the relatively simple nature of the data analysis involving primarily the tabulation of responses, the discussion section appears excessively lengthy. Additionally, the discussion seems to rely heavily on existing literature rather than focusing on the implications of the current findings. The discussion should be centered around the results of this study. It is recommended to concisely summarize the most significant findings from the data analysis.
Furthermore, after revising the discussion, the conclusion should briefly address the potential clinical applications of this research and its potential contributions to nursing practice and patient care.
Page 9. Line 265-299 The discussion summarizes the most significant findings
Page 9. Line 30-06 conclusion rephrased to address the potential clinical applications and contribution.
Comment 21. Literature citations
Please place all reference numbers within the period at the end of the sentence. For example, "This finding is consistent with previous research [1]."
Corrected throughout the document